# Effects of Water Stress and Auxin Application on Growth and Yield of Two Sugarcane Cultivars under Greenhouse Conditions

**Jiraporn Bamrungrai [1], Brenda Tubana [2], Vidhaya Tre-loges [3], Arunee Promkhambut [4] and Anan Polthanee [1,\*]**

[1] Department of Agronomy, Faculty of Agriculture, Khon Kaen University, Khon Kaen 40002, Thailand; Jira.bamrungrai@gmail.com

[2] School of Plant, Environmental, and Soil Sciences, Louisiana State University, 309 Sturgis Hall, Baton Rouge, LA 70803, USA; btubana@agcenter.lsu.edu

[3] Department of Soil Science and Environment, Faculty of Agriculture, Khon Kaen University, Khon Kaen 40002, Thailand; Vidtre1@kku.ac.th

[4] Department of Agricultural Extension and Agricultural Systems, Faculty of Agriculture, Khon Kaen University, Khon Kaen 40002, Thailand; arunee@kku.ac.th

\* Correspondence: panan@kku.ac.th; Tel.: +66-81-047-7800

**Abstract:** Water stress (drought and flood) is one of the major factors that limits sugarcane growth and yield. The two greenhouse experiments were conducted at Khon Kaen University, Thailand. The first experiment investigated the individual and combined effects of droughts and floods on two sugarcane cultivars. The results showed that photosynthetic potential, based on chlorophyll *a* fluorescence (PSII) and chlorophyll content, exhibited a response to the water regime treatments. However, stomatal conductance in the K93-219 cultivar was higher than the KK3 cultivar. Similarly, plant height, number of tillers, number of stems, fresh stem weight, and sugar quality were not affected by the varying water regime conditions imposed on both of the sugarcane cultivars. However, drought or flood conditions, whether alone or in combination, reduced the fresh stem weight, with regards to the water regimes and cultivars. In general, a combination of drought and flood reduced the fresh stem weight as opposed to drought or flood alone. The KK3 cultivar gave a higher fresh stem weight than the K93-219 cultivar under dual stress conditions. The second experiment investigated the auxin application rates at different growth stages on two sugarcane cultivars under flood conditions. The results revealed that the application of auxin at 10 mg L$^{-1}$ increased the number, and fresh weight, of adventitious roots over the control (0 mg L$^{-1}$). The cultivar K93-219 produced a higher number, and fresh weight, of adventitious roots. However, the amount of aerenchyma in the adventitious roots was not affected by auxin application rates, growth stages, or cultivars. Similarly, plant height, leaf width, number of stems, fresh stem weight, and sugar quality were not affected by auxin rates. Auxin application at five months of age increased leaf width and fresh stem weight over the control. The cultivar K93-219 tended to produce a higher fresh stem weight than the KK3 cultivar.

**Keywords:** drought; flooding; physiological trait; adventitious root; sugar quality

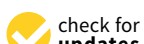



## 1. Introduction

The largest sugarcane (*Saccharum officinarum* L.) production areas in Thailand are located in the northeast. Sugarcane is usually planted in upland fields. Due to drought, rice (*Oryza sativa* L.) production was low, which led farmers to shift from rice to sugarcane production, especially in the upper paddy fields along the watershed landform. Most of the farmers in the northeast of Thailand plant sugarcane during the late rainy season (October–November). After planting, a long dry season usually follows which lasts for approximately six months (November–April), triggering drought during the vegetative growth stage of sugarcane. Such water stress conditions during the vegetative growth stage of sugarcane are further compounded by flooding during the late growth stage due to intense rainfall in the mid-rainy season. Hence, drought and flood conditions limit the growth and productivity of sugarcane in these areas.

Drought is one of the limiting factors that reduce production for many crops [1]. The amount of water used by a crop is closely associated with photosynthetic activity, dry matter production, and yield in many species [2,3]. The tillering and grand growth stages, known as the sugarcane formative phase, have been identified as the critical periods for sufficient water supply [4], mainly because this is the phase when 70–80% of cane yield is produced [5]. Morphological alterations, including reductions in leaf area, root growth, and stomatal closure, are symptoms of plants suffering from drought [6–8]. Drought during the formative phase in sugarcane reduced dry matter from 46% to 61% [4]. Sugarcane cultivars differ in their ability to withstand drought stress, thus, the level of yield losses differs as well [4,9]. Sugarcane plant tolerance showed higher photosynthetic activity compared to susceptible plants [10]. In general, drought decreases the photochemical efficiency (PSII) and the ability of the cultivar to maintain a high level of *Fv/Fm*, indicative of the radiation use efficiency and carbon assimilation, which has become a promising tool to select cultivars more tolerant to drought [11]. Gene regulation in sugarcane leaves under drought stress, such as aquaporin, late embryogenesis abundant proteins, auxin-related proteins, transcription factors, heat shock proteins, light-harvesting chlorophyll a-b binding proteins, disease resistance proteins, and ribosomal proteins which contribute to the sugarcane breeding program, was reported by [12]. The drought stress response is essentially driven by phytohormones, namely abscisic acid, auxin, gibberellic acid, cytokinin, brassinosteroid, jasmonic acid, ethylene, and strigolactone, as reported by [13]. Similarly, efficiently tolerated freezing stress (cold-induced water stress) through sufficient osmotic adjustment ability and antioxidant system provides promising avenues for genetic engineering of cold-tolerance in crop plants [14]. Flooding caused the deficiency of oxygen in the soil environment and resulted in inhibiting water and nutrient uptake. Flooding induced severe deficiencies of N, P, and K in sugarcane; the N and K concentration were below the critical deficiency level [15]. Flooding also reduces the photosynthetic rate of many plant species [16,17]. The decline in photosynthesis under flooding has been attributed to stomatal closure [16], reduction in leaf chlorophyll content [16], ethylene production, reduced sink demand, and disruption in photosynthate transport. Overlapping responses to drought and flooding included positive regulation of trehalose and sucrose metabolism and negative regulation of cellulose tubulin, photosystem II and I, and chlorophyll biosynthesis [18]. In sugarcane, flood conditions impair the production of the tiller and the elongation of established tillers; the longer the duration of flooding the higher the reduction [19]. The decline of plant growth when exposed to flooding stress was dependent on the phases of growth of the plants based on when the plants were flooded and the length of flooding duration. Flooding stress can decrease the physiology process in susceptible plants, both in the vegetative and the generative phases [20]. Flooding over 15–60 days over the grand growth phase decreased the yield of sugarcane by approximately 5–30% because of the lack of nutrition and water uptake [21], while three months of flooding decreased the yield by 18–37% in the plant cane and 61–63% in the second ratoon [22]. In addition, a reduction in sugarcane yield by 14–50% was reported by [23]. Sugarcane cultivars have a different capability in adapting to flood conditions [24–26]. For sugarcane, adventitious roots were highlighted as a common response in flood-tolerant species [27]. These adventitious roots have high porosity, help plants to survive under flood conditions, and replace, in some way, the functions of the older root system [28]. Most reports of adventitious root formation under flood conditions were induced by ethylene accumulation in the plants [29–32]. In addition, the application of auxin induced the formation of adventitious roots in non-flooded plants [33]. The application of a high concentration of auxin increases ethylene production in many plants [34,35], which stimulates hypertrophy and adventitious root formation at the base of the stem just above the waterline [35]. Auxin IBA combined with NAA is a positive regulator of sugarcane microshoot adventitious root formation [36]. The exogenous application of gibberellic acid (GA$_3$) by priming on seeds of common wheat varieties with different vernalization (low temperature) and photoperiod requirements affected the transcription from the vegetative to the generative stage, as reported by [37].

The purpose of this research was (i) to investigate the effect of drought and flood on the growth and the sugar quality of two sugarcane cultivars, (ii) to evaluate the physiological characters of two sugarcane cultivars in response to drought and flood conditions, and (iii) to investigate the effects of auxin application rates at different growth stages on adventitious root development, aerenchyma formation, growth, and sugar quality of two sugarcane cultivars grown under flooding stress.

## 2. Materials and Methods

The experiments were conducted from December 2015 to December 2016 (Experiment 1) and from December 2017 to December 2018 (Experiment 2) in the greenhouse at the Department of Plant Science and Agricultural Resources, Faculty of Agriculture, Khon Kaen University, Thailand (16°46′ N, 102°80′ E). In Experiment 1, the treatment structure was a 2 × 9 factorial arranged in a randomized complete block design (RCBD) with three replications. The first factor was two sugarcane cultivars; KK3 and K93-219. The second factor consisted of nine water regimes. Soil moisture was maintained at field capacity during the entire growing period (control), drought at 3 months of age (MOA) for 30 days (D1), drought at 3 MOA for 60 days (D2), flood at 7 MOA for 30 days (F1), flood at 7 MOA for 60 days (F2), drought at 3 MOA for 30 days combined with flood at 7 MOA for 30 days (D1 + F1), drought at 3 MOA for 30 days combined with flood at 7 MOA for 60 days (D1 + F2), drought at 3 MOA for 60 days combined with flood at 7 MOA for 30 days (D2 + F1), and drought at 3 MOA for 60 days combined with flood at 7 MOA for 60 days (D2 + F2). Drought was designed as irrigation at 50% of field capacity (FC), while flood was designed as maintaining the water level at 10 cm above the soil surface. In the present experiment, the soil moisture level designed in the drought period is relatively high. This was to simulate sugarcane grown in lowland fields where there has been a shift from growing rice. Water regime treatment exclusions are shown in Figure 1.

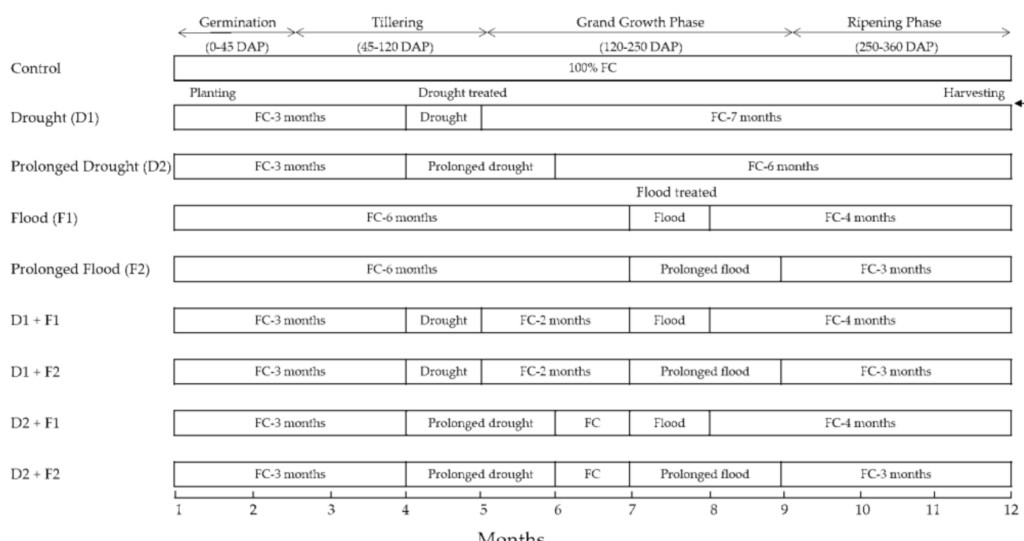

**Figure 1.** Period of drought and flood exclusion treatments, drought = 50% of FC, flood = maintaining water level at 10 cm above soil surface.

In Experiment 2, treatments included two sugarcane cultivars (KK3 and K93-219), four auxin rates (0, 10, 20, and 30 mg L$^{-1}$), and auxin application at three growth stages (4, 5, and 7 months of age). The treatments (2 × 4 × 3 factorial structure) were arranged in a randomized complete block design with three replications. Auxin application was initiated at 7 days after flood treatment (DAT) and consequently at 14 and 21 DAT during the flooding for 30 days of each growth stage. Soil moisture level was maintained at FC before and after the flood treatment in this study.

### 2.1. Cultural Detailed

Cultural methods were similar in the two pot experiments. Soil samples were taken at 0–30 cm depth from the farmer's upper paddy field located in Muang district, Khon Kaen province. The soil was air-dried, homogenized, and passed through a 2 mm sieve. Pots (60 cm diameter × 50 cm height) were filled with 100 kg of processed soil. A nursery was established in which sugarcane seedlings were raised from single-budded cane chips. The seedlings were allowed to grow for 30 days and then transferred to the pots. Fertilizer grade 15-15-15 (N, $P_2O_5$, $K_2O$) was applied at the rate of 312 kg ha$^{-1}$ (18.48 g pot$^{-1}$) in a split application at the planting and the tillering stages. The pests were controlled as necessary.

### 2.2. Measurement

2.2.1. Soil Property

The soil used in both experiments had a loam texture. The soil was slightly acidic with a pH of 5.77 and a low organic matter content (0.3%). The total N, P, and K measured from the soil were 145, 6.8, and 584 mg kg$^{-1}$, respectively. The soil moisture content at field capacity (FC) and permanent wilting point (PWP) were 12.14% and 4.03% by weight, respectively.

2.2.2. Physiological Characters

In Experiment 1, stomatal conductance was recorded three times from the third or fourth leaves from the top of the plant's main stem between 10:00 am and 1:00 pm under a clear sky at 1, 2, 4, and 8 days after drought treatment at 4 MOA and flood treatment at 7 MOA. Stomatal conductance was measured using the SC-1 Leaf Porometer (Decagon Devices, Inc., Washington, DC, USA). Furthermore, chlorophyll *a* fluorescence and chlorophyll content were measured three times from the third or fourth leaves from the top of the plant's main stem between 10:00 am and 1:00 pm under a clear sky at 3, 5, 7, 9 MOA, and at harvest. Chlorophyll *a* fluorescence was measured using a portable chlorophyll fluorescence meter (Mini-pam, Watz, Germany). Chlorophyll content was measured by using SPAD chlorophyll meter reading (SPAD502-Plus, Komica Minolta, Tokyo Japan). In Experiment 2, green leaves were counted (leaves that were over 50% green) at harvest.

2.2.3. Root Growth

The number of adventitious roots were counted at 3 and 7 days after the initiation of flood treatment in both experiments. Adventitious roots were sampled on the last day of flood treatment by cutting them from the plants and their fresh and dry weights were measured in both experiments. The dry weight of the original roots was also recorded at harvest in Experiment 1.

2.2.4. Aerenchyma Formation

In general, flooded plants produced three types of adventitious roots after flooding: (i) roots initiating from aerial nodes, (ii) roots initiating from the pre-existing primordial roots and developed under water, and (iii) secondary roots initiating from the newly developed roots [32,38]. In Experiment 2, adventitious roots initiated from the pre-existing primordial roots and developed under water were evaluated for the presence of aerenchyma, compared with auxin application rates, growth stages, and cultivars. In addition, aerenchyma formation in adventitious roots was compared with original roots in this study. Cross-sections of fresh roots exhibiting a thickness of 80–100 μm were made every 5 cm starting at 0.5 cm beyond the root base (root–shoot junction) [39]. A microtome (Plant Microtome Automatic MT-3, NK System) was used to prepare the cross-sections. The cross-sections of fresh roots were fixed in stained 70% FAA (formalin, acetic acid, 70% ethanol; 1:1:18) followed by [40] and stained with toluidine blue (0.01%) for 3 min. The amount of aerenchyma in the root cortex was visually scored: 0 (no aerenchyma), 0.5 (par-

tial formation), 1 (radial formation), 2 (radial formation extended toward epidermis), and 3 (well-formed aerenchyma) [39].

### 2.2.5. Shoot Growth and Sugar Quality

In Experiment 1, the tiller number was recorded at 3, 5, and 9 months of age (MOA). Plant height was measured from the base (soil surface) to the leaf tip at 3, 5, 9 MOA, and at harvest. The number of stems and fresh stem weight were recorded at harvest. In Experiment 2, plant height, leaf width (the fully expanded leaves), number of stems, and fresh stem weight were measured at harvest. Sugarcane juice was extracted from each stalk and analyzed for Brix (%), polarity (%), fiber (%), purity (%), and commercial cane sugar (CCS, %) in both experiments.

### 2.3. Statistical Analysis

Data for each experiment were subjected to individual analysis of variance (ANOVA). The data were analyzed using the computer software Statistix 10. The analysis of variance for all measured variables was performed to test for the significance of the treatment effect. For variables with significant treatment effect detected ($p \leq 0.05$) the comparison of treatment means was performed using the least significant difference (LSD) test at $p \leq 0.05$ level of confidence.

## 3. Results

**Experiment 1.** *The effects of drought and flood at different growth stages on morphological and physiological characters, growth, and sugar quality of two sugarcane cultivars under greenhouse conditions.*

### 3.1. Physiological Characters Performance
### 3.1.1. Stomatal Conductance

Stomatal conductance is a measure of the ability of stomata for gas exchange of $CO_2$, $H_2O$ vapor, and other gas derived from the atmosphere [41]. In this study, in the KK3 cultivar, the stomatal conductance was significantly different based on the water regimes imposed on the sugarcane at 1 and 8 days after treatment (DAT) but there was no significant difference at 2 and 4 DAT from the control plants (Table 1). This indicates that a decreased soil water content during drought shows slight changes in the stomatal conductance of sugarcane. This is in agreement with the findings of a previous study, which illustrates that the stomatal conductance of sugarcane that experienced moderate drought stress differed significantly from the control plants [42–44]. Maximum stomatal conductance was observed in prolonged flooding stress (F2) in both 1 and 8 DAT. For the K93-219 cultivar, the stomatal conductance was significantly different based on the water regimes at 2 and 8 DAT (Table 1). The highest stomatal conductance was observed in prolonged drought combined with flooding stress observed in the control plant at 8 DAT. In contrast, the flooded plant had a higher stomatal conductance than that of the control plants as reported by [25]. Plants that experienced flooding stress also showed an increased number of stomata [1]. The KK3 cultivars showed higher stomatal conductance values than the K93-219 cultivars in this study (Table 1). Previous studies [10,45] reported that sugarcane cultivars differed in stomatal conductance under drought stress. In general, the tolerant cultivars showed significantly higher stomatal conductance than that of susceptible cultivars [10]. The different stomatal conductivity values from low to high of the various sugarcane cultivars in response to flood conditions were reported by [26]. In general, the tolerant cultivar group exhibited higher stomatal conductance than those of the susceptible cultivar group.

**Table 1.** Stomatal conductance (gs) of two sugarcane cultivars at 1, 2, 4, and 8 days after initiation of treatment (DAT) and harvest as affected by drought or flood treatment, Experiment 1.

| Cultivar | Water Regime | Stomatal Conductance ($\mu$mol/m$^2$/s) | | | |
|---|---|---|---|---|---|
| | | **1 DAT** | **2 DAT** | **4 DAT** | **8 DAT** |
| | Control | 274.5ab | 342.2 | 299.73 | 285.8ab |
| | Drought (D1) | 114.7c | 234.3 | 237.94 | 195.2abc |
| | Prolonged Drought (D2) | 222.4bc | 474.9 | 228.75 | 286.4a |
| | Flood (F1) | 288.5a | 210.4 | 282 | 255.6ab |
| | Prolonged Flood (F2) | 167.2c | 182.9 | 177.43 | 176.4bc |
| KK3 | D1 + F1 | 169.4c | 155.8 | 119.2 | 219.7abc |
| | D1 + F2 | 241.6ab | 267.2 | 285.36 | 253.6ab |
| | D2 + F1 | 246.0ab | 279.9 | 248.72 | 147.1c |
| | D2 + F2 | 173.0c | 266.9 | 336.72 | 170.1bc |
| | F-test | * | ns | ns | * |
| | CV (%) | 18.9 | 36.7 | 30.1 | 21.62 |
| | Control | 269.1 | 309.8ab | 229.7 | 350.4a |
| | Drought (D1) | 152.7 | 195.3abc | 219.1 | 200.0c |
| | Prolonged Drought (D2) | 200.4 | 295.5ab | 230.0 | 166.9c |
| | Flood (F1) | 182.5 | 288.6ab | 345.6 | 250.0bc |
| | Prolonged Flood (F2) | 136.7 | 162.3c | 152.4 | 172.7c |
| K93-219 | D1 + F1 | 149.8 | 186.9bc | 210.4 | 179.9c |
| | D1 + F2 | 207.9 | 341.2a | 216.0 | 189.4c |
| | D2 + F1 | 263.7 | 314.07ab | 287.4 | 299.5ab |
| | D2 + F2 | 220.0 | 195.3abc | 246.0 | 151.6c |
| | F-test | Ns | * | ns | * |
| | CV (%) | 32 | 15.1 | 24.82 | 20.7 |

Notes: ns, * = non-significant and significant at $p \leq 0.05$ probability levels, respectively. Mean with the difference, small letters in each column are significantly different by least significant difference ($p \leq 0.05$); drought treated at 4 MOA, flood treatment at 7 MOA.

### 3.1.2. SPAD Chlorophyll Meter Reading (SCMR)

Leaf chlorophyll content was estimated using SCMR. This index was preferred because of the strong relationship between the readings of the portable chlorophyll meter and the leaf chlorophyll content [11,46,47]. In the present study, water regime treatments did not show a significant difference in SCMR across cultivars at 3, 5, 7, and 9 months of age, and at harvest (Table 2). This indicates that the plants exposed to drought or flood stress alone or a combination of both (drought followed by flood) at any given growth stage were unaffected. Drought conditions (15% *v/v* soil moisture), as reported by [26], reduced SCMR, but was unaffected at a fairly level flood and prolonged flood alone, combination flood followed by prolonged drought, and prolonged flood followed by prolonged drought.

**Table 2.** SPAD chlorophyll meter reading (SCMR) of two sugarcane cultivars at 3, 5, 7, 9 months of age, and at harvest as affected by drought, flood, and combined treatment, Experiment 1.

| Cultivar | Water Regime | SCMR | | | | |
|---|---|---|---|---|---|---|
| | | **3 MOA** | **5 MOA** | **7 MOA** | **9 MOA** | **Harvest** |
| | Control | 43.2 | 61.5 | 74.1 | 67.8 | 32.6 |
| | Drought (D1) | 45.5 | 52.6 | 66.3 | 59.5 | 22.3 |
| | Prolonged Drought (D2) | 47.4 | 78.4 | 80.2 | 79.3 | 24.0 |
| | Flood (F1) | 44.0 | 71.7 | 59.0 | 65.3 | 26.5 |
| KK3 | Prolonged Flood (F2) | 48.2 | 52.9 | 69.4 | 61.2 | 31.9 |
| | D1 + F1 | 43.0 | 32.4 | 44.0 | 38.2 | 32.7 |
| | D1 + F2 | 47.5 | 92.9 | 81.1 | 87.0 | 23.3 |
| | D2 + F1 | 47.9 | 58.4 | 65.9 | 62.2 | 32.0 |
| | D2 + F2 | 43.5 | 71.9 | 82.5 | 77.2 | 28.1 |

**Table 2.** *Cont.*

| Cultivar | Water Regime | SCMR | | | | |
|---|---|---|---|---|---|---|
| | | 3 MOA | 5 MOA | 7 MOA | 9 MOA | Harvest |
| | F-test | Ns | ns | ns | Ns | ns |
| | CV (%) | 12.9 | 27.5 | 30.3 | 26.3 | 21.3 |
| | Control | 55.1 | 92.1 | 87.0 | 89.5 | 60.6 |
| | Drought (D1) | 42.8 | 74.0 | 91.8 | 82.9 | 32.1 |
| | Prolonged Drought (D2) | 45.5 | 65.0 | 76.3 | 70.6 | 68.5 |
| | Flood (F1) | 51.4 | 85.3 | 88.3 | 86.7 | 35.4 |
| K93-219 | Prolonged Flood (F2) | 49.4 | 94.1 | 85.3 | 89.7 | 32.9 |
| | D1 + F1 | 47.3 | 88.3 | 92.9 | 90.6 | 31.5 |
| | D1 + F2 | 50.1 | 68.8 | 85.8 | 77.2 | 33.9 |
| | D2 + F1 | 48.6 | 83.8 | 76.3 | 80.5 | 34.8 |
| | D2 + F2 | 44.5 | 90.6 | 88.5 | 89.5 | 49.5 |
| | F-test | Ns | ns | ns | Ns | ns |
| | CV (%) | 10.8 | 12.7 | 11.3 | 9 | 5.29 |

Notes: ns = non-significant at $p \leq 0.05$; MOA = month of age, drought treated at 4 MOA, flood treated at 7 MOA.

### 3.1.3. Chlorophyll a Fluorescence

Photosystem II (PSII) is the component of the photosynthetic complex having an essential role in response to environmental stress. The PSII can be accessed via the variable-to-maximum chlorophyll a fluorescence ratio (*Fv/Fm*) [11,48]. In the present study, the KK3 cultivar water regimes had no significant effect on the chlorophyll *a* fluorescence ratio at 3, 5, 7, 9 MOA, and at harvest (Table 3). For the K93-219 cultivar, the *Fv/Fm* ratio values were significantly different for the effect on chlorophyll *a* fluorescence at 3 MOA but there was no significant difference at 5, 7, 9 MOA, and harvest (Table 3). The highest chlorophyll *a* fluorescence at 3 MOA was observed in the control plant. This indicates that the plants exposed to drought or flood alone or a combination of treatments (drought followed by flood) at any given growth stage, and cultivars were unaffected in terms of their photochemical efficiency. It was reported by [49] that the *Fv/Fm* values of sugarcane were not significantly different between well-watered (70 to 90% of field capacity), drought (20–40% of field capacity), and across cultivars. A reduction in photosynthesis (9%) in the flooded plants in comparison with the control was reported by [38].

**Table 3.** Chlorophyll a fluorescence ratio (*Fv/Fm*) of two sugarcane cultivars at 3, 5, 7, 9 months of age, and harvest as affected by drought, flood, and combined treatment, Experiment 1.

| Cultivar | Water Regime | Chlorophyll *a* Fluorescence Ratio (*Fv/Fm*) | | | | |
|---|---|---|---|---|---|---|
| | | 3 MOA | 5 MOA | 7 MOA | 9 MOA | Harvest |
| | Control | 689.6 | 792.7 | 722.5 | 819 | 780.1 |
| | Drought (D1) | 632.6 | 794.6 | 738.0 | 813.3 | 779.0 |
| | Prolonged Drought (D2) | 682.6 | 802.3 | 691.0 | 799.6 | 789.3 |
| | Flood (F1) | 709.1 | 798.3 | 746.6 | 789.9 | 779.4 |
| KK3 | Prolonged Flood (F2) | 701.0 | 772.3 | 743.6 | 745.6 | 782.0 |
| | D1 + F1 | 695.3 | 788.0 | 725.8 | 798.4 | 798.2 |
| | D1 + F2 | 673.6 | 782.6 | 756.6 | 804.3 | 752.6 |
| | D2 + F1 | 681.3 | 778.3 | 727.6 | 776.6 | 769.6 |
| | D2 + F2 | 670.6 | 768.6 | 744.0 | 808.3 | 775.3 |
| | F-test | ns | Ns | ns | ns | ns |
| | CV (%) | 7.7 | 3.22 | 4.66 | 3.31 | 3.23 |
| | Control | 750.3a | 803.3 | 742.7 | 798.1 | 808.0 |
| | Drought (D1) | 743.4ab | 737.6 | 824.5 | 836.3 | 818.6 |
| K93-219 | Prolonged Drought (D2) | 584.0d | 780.7 | 734.9 | 723.4 | 748.8 |
| | Flood (F1) | 737.3abc | 796.3 | 752.0 | 764.3 | 763.3 |
| | Prolonged Flood (F2) | 684.6bc | 786.3 | 739.0 | 788.0 | 770.0 |

**Table 3.** *Cont.*

| Cultivar | Water Regime | Chlorophyll *a* Fluorescence Ratio (*Fv/Fm*) | | | | |
|---|---|---|---|---|---|---|
| | | 3 MOA | 5 MOA | 7 MOA | 9 MOA | Harvest |
| K93-219 | D1 + F1 | 671.8c | 774.3 | 737.2 | 783.1 | 778.5 |
| | D1 + F2 | 698.6abc | 773.5 | 745.8 | 824.1 | 769.4 |
| | D2 + F1 | 700.3abc | 761.6 | 753.0 | 809.7 | 793.3 |
| | D2 + F2 | 741.6abc | 801.9 | 811.4 | 801.7 | 786.3 |
| | F-test | * | Ns | ns | ns | ns |
| | CV (%) | 4.01 | 4.46 | 5.24 | 3.16 | 2.8 |

Notes: ns, * = non-significant and significant at $p \leq 0.05$ probability levels, respectively; MOA = month of age, drought treated at 4 MOA, flood treated at 7 MOA.

### 3.2. Root Growth and Development

Flooded plants produced three types of adventitious roots after flooding in the present study (Figure 2). The first type of adventitious root appeared from the nodes under the water. A second type of root developed from the first type which grew upward against gravity. A third type of root emerged at the aerial nodes under prolonged flooding. This finding is in agreement with a previous study by [32]. In the present study, water regimes of flooding treatments had no significant effect on the number of adventitious roots in either the KK3 or K93-219 cultivars at 3 and 7 DAT (Table 4). Similarly, flooding treatments did not exhibit any significant difference in fresh adventitious root weight and dry weight of either sugarcane cultivars (Table 4). However, K93-219 tended to give higher fresh and dry adventitious root weights than that of the KK3 cultivar. Adventitious root development is a mechanism of sugarcane to adapt to flooding stress as reported by several previous studies [23,26]. These roots are adapted to flooding conditions, as opposed to the original roots, because they have much larger intercellular spaces [50]. Similarly, the two sugarcane cultivars did not exhibit significant differences in fresh and dry adventitious root weights, including the number of roots per plant across the water regimes in this study (Table 4). In general, cultivars that have a tolerance to flooding stress produced larger numbers of adventitious roots (bushy and long) than that of the susceptible cultivars [1]. In the case of original root weight (root in soil), water regimes, including the control, had a significant effect on fresh and dry original root weights (Table 4). All water regime treatments reduced the fresh and dry original root weights compared to the control, with regards to water regimes in both cultivars. In the previous study, the dry original root weight decreased in flooded plants, including those exposed to a combination of treatments (i.e., flood = 24.0%, prolonged flood = 25.3%, flood + prolonged drought = 20.4% and prolonged flood + prolonged drought = 25.3%) [32]. The greatest reduction of fresh original root weight was observed in the drought combined with flood treatment from 59.6 to 65.8% in the KK3 and K93-219 cultivars, respectively.

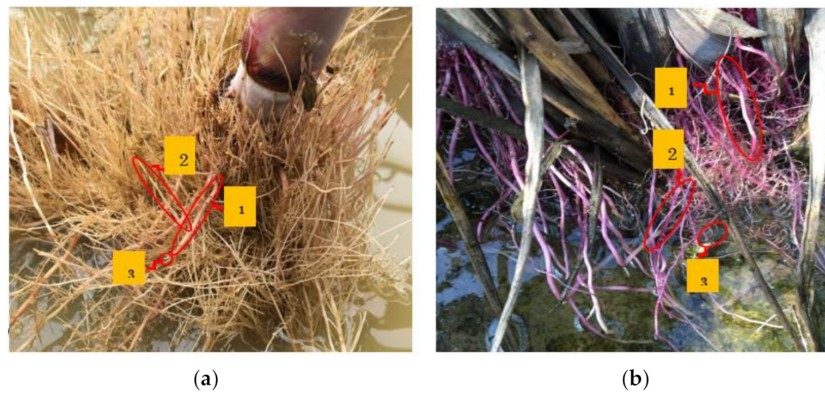

(a)       (b)

**Figure 2.** The adventitious root of the KK3 (**a**) and K93-219 (**b**) cultivars separated into 3 types: primary (1), secondary (2), and roots initiated from secondary adventitious roots (3).

**Table 4.** Adventitious root number and adventitious fresh and dry root weight of sugarcane at 3 and 7 days after treated (DAT), and fresh and dry original root weight at harvest as affected by drought, flood, and combined treatments. Experiment 1.

| Cultivar | Water Regime | Adventitious Root (No. Plant$^{-1}$) | | Adventitious Root Weight (g) | | Original Root Weight (g) | |
|---|---|---|---|---|---|---|---|
| | | 3 DAT | 7 DAT | Fresh Weight | Dry Weight | Fresh Weight | Dry Weight |
| KK3 | Control | 0 | 0 | 0 | 0 | 78.7a | 19.4a |
| | Drought (D1) | 0 | 0 | 0 | 0 | 40.3c | 12.5c |
| | Prolonged Drought (D2) | 0 | 0 | 0 | 0 | 34.2cd | 16.1b |
| | Flood (F1) | 20 | 60 | 28.0 | 7.4 | 64.2bc | 18.7ab |
| | Prolonged Flood (F2) | 15 | 56 | 36.0 | 6.9 | 42.6c | 13.9c |
| | D1 + F1 | 17 | 54 | 32.2 | 7.4 | 26.9d | 10.5d |
| | D1 + F2 | 19 | 46 | 31.8 | 8.3 | 54.6bc | 12.1c |
| | D2 + F1 | 18 | 48 | 30.5 | 8.7 | 57.9bc | 12.9c |
| | D2 + F2 | 21 | 48 | 34.2 | 8.9 | 66.3bc | 13.3c |
| | F-test | ns | ns | ns | Ns | ** | * |
| | CV (%) | 8.4 | 12.9 | 5.9 | 15 | 18.5 | 15.9 |
| K93-219 | Control | 0 | 0 | 0 | 0 | 63.6a | 15.8a |
| | Drought (D1) | 0 | 0 | 0 | 0 | 36.2bc | 12.6cd |
| | Prolonged Drought (D2) | 0 | 0 | 0 | 0 | 30.9c | 10.6d |
| | Flood (F1) | 23 | 60 | 47.3 | 8.8 | 41.3bc | 13.4c |
| | Prolonged Flood (F2) | 19 | 63 | 51.1 | 10.2 | 67.6a | 15.9a |
| | D1 + F1 | 21 | 57 | 49.2 | 10.9 | 25.7d | 10.8d |
| | D1 + F2 | 22 | 51 | 45.7 | 9.4 | 34.3c | 10.5d |
| | D2 + F1 | 17 | 48 | 48.9 | 9.5 | 49.5b | 14.4bc |
| | D2 + F2 | 21 | 54 | 52.2 | 9.1 | 53.6ab | 14.8b |
| | F-test | ns | ns | ns | Ns | * | * |
| | CV (%) | 8.4 | 16.5 | 5.9 | 14.2 | 15.8 | 17.49 |

Notes: ns, **, * = non-significant and significant at $p \leq 0.01$ and $p \leq 0.05$ probability levels, respectively. Mean with the difference, small letters in each column are significantly different by least significant difference ($p \leq 0.05$); drought treated at 4 MOA, flood treated at 7 MOA, DAT = day after treated.

### 3.3. Shoot Growth and Yield

The water regimes had no significant ($p \leq 0.05$) effect on the plant height and tiller numbers at 3, 5, and 9 MOA, and at harvest for the KK3 cultivar (Table 5). Regarding the number of stems and fresh stem weight, water regimes had a significant effect on the fresh stem weight, but there was no significant difference in the number of stems (Table 5). The lowest fresh stem weight was obtained in the drought combined with prolonged flood treatment, a 57% reduction compared to the control. However, all of the water stress treatments reduced fresh stem weight from between 19.1% and 57.1% as compared to the control. For the K93-219 cultivar, water regimes had no significant effect on the plant height at 3, 5, and 9 MOA, but there was a significant difference in plant height at harvest. The highest plant height was observed in drought combined with flood treatment (D1 + F1) (Table 5). The water regimes showed no significant difference on the tiller numbers at 3 and 5 MOA but showed a significant difference at 9 MOA. The highest tiller number was observed in prolonged drought combined with prolonged flood treatment (D2 + F2). The water regimes showed no significant difference in the number of stems but showed a significant difference in fresh stem weight at harvesting time. The lowest fresh stem weight was observed in the drought combined with flood treatment (D1 + F1), showing a 61% reduction compared to the control. However, all of the water stress treatments reduced the fresh stem weight by 10% to 60% as compared to the control. This was associated with the dual stress which had a significant effect on reducing the original root growth (root in soil) as mentioned above and resulted in the low uptake of water and nutrients. In comparison with drought and flooding stress alone, the yield (fresh stem weight) was reduced due to drought and flooding stress (average from drought and flood alone), by 26% and 14%, respectively, in reference to the control for the KK3 cultivar. In contrast, the yield was reduced due to drought and flooding stress (average from drought and flood alone) by 45% and 13%, respectively, in reference to the control for the K93-219 cultivar. This indicates that sugarcane is more sensitive to drought stress than flooding stress. Drought can lead to

yield losses ranging from 46.2% to 50% [51,52]. Sugarcane yield reduced by 14–50% when they experienced flooding [21] and by 18–37% compared to the well-watered control [24].

**Table 5.** Shoot growth parameters of two sugarcane cultivars as affected by drought, flood, and combined treatment, Experiment 1.

| Cultivar | Water Regime | Tiller (No. Plant⁻¹) | | | Stem Number (No. Plant⁻¹) | Plant Height (cm) | | | | Stem Fresh Weight |
|---|---|---|---|---|---|---|---|---|---|---|
| | | 3 MOA | 5 MOA | 9 MOA | Harvest | 3 MOA | 5 MOA | 9 MOA | Harvest | (kg Plant⁻¹) |
| KK3 | Control | 4.1 | 4.7 | 5.2 | 2.2 | 144.3 | 147.3 | 195.0 | 310.0 | 2.1a |
| | Drought (D1) | 4.6 | 8.1 | 11.0 | 3.3 | 149.3 | 135.1 | 199.6 | 278.4 | 1.4bc |
| | Prolonged Drought (D2) | 3.6 | 7.0 | 10.0 | 3.6 | 152.0 | 138.6 | 220.3 | 256.6 | 1.7ab |
| | Flood (F1) | 3.1 | 2.7 | 4.3 | 2.8 | 126.0 | 117.3 | 216.7 | 263.7 | 1.7ab |
| | Prolonged Flood (F2) | 3.3 | 4.0 | 4.6 | 3.6 | 137.0 | 153.0 | 202.6 | 280.0 | 1.9ab |
| | D1 + F1 | 4.3 | 4.6 | 4.8 | 4.1 | 145.7 | 142.5 | 214.7 | 238.4 | 1.3bc |
| | D1 + F2 | 4.3 | 6.3 | 8.3 | 2.3 | 127.0 | 135.6 | 192.6 | 310.0 | 0.9c |
| | D2 + F1 | 5.0 | 5.5 | 6.0 | 3.3 | 162.3 | 160.6 | 184.6 | 300.0 | 1.2bc |
| | D2 + F2 | 5.3 | 6.8 | 8.3 | 2.3 | 146.3 | 140.0 | 211.3 | 296.6 | 1.6ab |
| | F-test | ns | ns | ns | ns | ns | ns | Ns | ns | * |
| | CV (%) | 39.1 | 33 | 41.26 | 30.5 | 9.0 | 9.0 | 15.1 | 11.3 | 10.9 |
| K93-219 | Control | 4.5 | 6.2 | 7.8c | 2.7 | 156.3 | 150.9 | 213.8 | 300.5a | 2.0a |
| | Drought (D1) | 6.0 | 7.2 | 8.3bc | 2.6 | 133.4 | 108.5 | 193.6 | 199.4b | 1.1b |
| | Prolonged Drought (D2) | 4.8 | 10.0 | 11.2ab | 2.6 | 158.2 | 157.0 | 207.6 | 169.4b | 1.1b |
| | Flood (F1) | 5.0 | 5.5 | 6.0c | 2.3 | 137.0 | 148.7 | 205.0 | 310.0a | 1.8ab |
| | Prolonged Flood (F2) | 5.3 | 7.3 | 9.3bc | 2.6 | 142.0 | 152.7 | 206.6 | 290.0a | 1.7ab |
| | D1 + F1 | 4.5 | 6.2 | 7.8c | 2.7 | 114.3 | 119.4 | 224.3 | 320.5a | 0.8b |
| | D1 + F2 | 5.3 | 7.8 | 10.2bc | 2.0 | 142.3 | 133.6 | 224.9 | 279.2a | 0.9b |
| | D2 + F1 | 4.3 | 6.8 | 9.3bc | 2.3 | 141.3 | 128.7 | 200.0 | 306.0a | 1.1b |
| | D2 + F2 | 6.3 | 11.0 | 14a | 3.2 | 134.3 | 162.8 | 208.1 | 291.5a | 1.1b |
| | F-test | ns | ns | * | ns | ns | ns | Ns | ** | * |
| | CV (%) | 32.1 | 19.39 | 22.35 | 22.48 | 14.9 | 11.6 | 4.2 | 6.4 | 25.4 |

Notes: ns, **, * = non-significant and significant at *p≤0.01* and *p* ≤ 0.05 probability levels, respectively. Mean with the difference small, letters in each column are significantly different by least significant difference (*p* ≤ 0.05); MOA = month of age, drought treated at 4 MOA, flood treated at 7 MOA.

### 3.4. Sugar Quality

In the present study, water regimes had no significant effect on sugar quality properties such as Brix, CCS, fiber, polarity, and purity for both cultivars at harvest (Table 6). The KK3 cultivar exposed to drought for 60 days tended to give the highest Brix content (25.4%) followed by the exposure to flooding for 30 days with a Brix content value of 24.9%. The K93-219 cultivar's highest Brix was observed in the control plant (25.2%) followed by the exposure to drought for 30 days (23.2%). In general, both sugarcane cultivars exhibited a high Brix content (ranging from 15.4% to 25.4%) although they were exposed to drought or flood alone, as well as to different combinations of these treatments. The Brix content of different sugarcane varieties, according to [26], ranges from17% to 19% under flooding conditions (flooding level at 10 cm above soil surface). In addition, sucrose (Brix as a solid substance) ranges from 21% to 23% under drought or flood alone or a combination of treatments, which was not significantly different from the control (22%) [32].

**Table 6.** Sugar quality of two sugarcane cultivars at harvest as affected by drought, flood, and combined treatments, Experiment 1.

| Cultivar | Water Regime | Brix (%) | Polarity (%) | Fiber (%) | CCS (%) | Purity (%) |
|---|---|---|---|---|---|---|
| KK3 | Control | 20.7 | 9.7 | 15.1 | 6.0 | 47.0 |
| | Drought (D1) | 21.9 | 13.5 | 16.6 | 7.3 | 62.6 |
| | Prolonged Drought (D2) | 25.4 | 16.1 | 15.5 | 8.8 | 64.1 |
| | Flood (F1) | 24.9 | 15.1 | 19.2 | 7.3 | 60.0 |
| | Prolonged Flood (F2) | 20.5 | 9.8 | 15.4 | 3.5 | 47.5 |

**Table 6.** *Cont.*

| Cultivar | Water Regime | Brix (%) | Polarity (%) | Fiber (%) | CCS (%) | Purity (%) |
|---|---|---|---|---|---|---|
| | D1 + F1 | 20.5 | 8.5 | 14.6 | 3.3 | 41.4 |
| | D1 + F2 | 18.4 | 8.3 | 16.1 | 5.5 | 41.0 |
| KK3 | D2 + F1 | 17.2 | 7.3 | 13.8 | 4.9 | 40.1 |
| | D2 + F2 | 17.2 | 6.8 | 15.2 | 1.4 | 39.3 |
| | F-test | ns | ns | ns | ns | ns |
| | CV (%) | 24.4 | 24.7 | 6.2 | 34.9 | 31.2 |
| | Control | 25.2 | 16.5 | 19.2 | 5.7 | 69.2ab |
| | Drought (D1) | 23.2 | 10.1 | 14.6 | 3.2 | 42.0b |
| | Prolonged Drought (D2) | 15.4 | 5.6 | 15.3 | 4.8 | 36.5b |
| | Flood (F1) | 17.9 | 6.8 | 16.8 | 2.9 | 37.3b |
| K93-219 | Prolonged Flood (F2) | 19.4 | 12.0 | 16.6 | 6.3 | 62.5ab |
| | D1 + F1 | 21.6 | 19.4 | 15.9 | 10.9 | 93.1a |
| | D1 + F2 | 19.6 | 9.8 | 20.0 | 5.3 | 49.4b |
| | D2 + F1 | 16.9 | 5.9 | 18.0 | 6.8 | 33.2bc |
| | D2 + F2 | 16.6 | 4.0 | 13.0 | 8.0 | 25.3c |
| | F-test | ns | ns | ns | ns | * |
| | CV (%) | 22.6 | 35.6 | 24.0 | 22.2 | 19.4 |

Notes: ns, * = non-significant and significant at $p \leq 0.05$ probability levels, respectively. Mean with the difference, small letters in each column are significantly different by least significant difference ($p \leq 0.05$) CCS = Commercial Cane Sugar.

**Experiment 2.** *The effects of auxin application rates at different growth stages on morphological, physiological, and anatomical characters of two sugarcane cultivars grown under flooding stress.*

### 3.5. Adventitious Root Development

The application of auxin induced the formation of adventitious roots in plants [33,53], due to the increase of ethylene production [34,35]. In the present study, auxin rates did not significantly affect the number of adventitious roots across the growth stages and cultivars at 3 DAT, but there was a significant difference observed at 7 DAT (Table 7). The application of auxin at 10 mg L$^{-1}$ resulted in the highest number of adventitious roots. For sugarcane cultivars, K93-219 produced a significantly higher number of adventitious roots across auxin rates and the growth stages at 3 and 7 DAT (Table 7). The external application of auxins inducing adventitious root development was reported in many previous studies [33,34,54] due to the stimulation of ethylene production [55].

**Table 7.** Adventitious root development of two sugarcane cultivars as affected by auxin application rates at different growth stages at the end of the flooding treatment period, Experiment 2.

| Treatment | AR Fresh Weight (g Plant$^{-1}$) | AR Dry Weight (g Plant$^{-1}$) | Number of Adventitious Roots | | Aerenchyma Score |
|---|---|---|---|---|---|
| | | | 3 DAT | 7 DAT | |
| Auxin Rates (A) | | | | | |
| 0 mg L$^{-1}$ | 15.1c | 4.8 | 38 | 86b | 2 |
| 10 mg L$^{-1}$ | 28.5a | 6.4 | 61 | 102a | 3 |
| 20 mg L$^{-1}$ | 23.9b | 5.9 | 50 | 88b | 3 |
| 30 mg L$^{-1}$ | 16.7c | 6.3 | 48 | 81c | 3 |
| F-test | * | ns | ns | ** | ns |
| Growth stages (G) | | | | | |
| 4 MOA | 24.4 | 6.2 | 44 | 89 | 3 |
| 5 MOA | 19.2 | 5.3 | 39 | 87 | 3 |
| 7 MOA | 20.3 | 6.2 | 55 | 90 | 3 |
| F-test | ns | ns | ns | ns | ns |
| Cultivars (C) | | | | | |
| KK3 | 22.7b | 6.1 | 39b | 80b | 3 |
| K93-219 | 32.4a | 7.28 | 52a | 97a | 3 |
| F-test | ** | ns | * | * | ns |

**Table 7.** *Cont.*

| Treatment | AR Fresh Weight (g Plant$^{-1}$) | AR Dry Weight (g Plant$^{-1}$) | Number of Adventitious Roots | | Aerenchyma Score |
|---|---|---|---|---|---|
| | | | **3 DAT** | **7 DAT** | |
| Interaction | | | | | |
| AxG | ns | ns | ns | ns | ns |
| AxC | * | ns | ns | * | ns |
| GxC | * | ns | ns | ns | ns |
| AxGxC | ns | ns | ns | ns | ns |

Notes: ns, **, * = non-significant and significant at $p \le 0.01$ and $p \le 0.05$ probability levels, respectively. Mean with the difference, small letters in each column are significantly different by least significant difference ($p \le 0.05$); AR = adventitious root, DAT = days after treated, MOA = month of age.

The application of auxin at different rates significantly affected fresh adventitious root weight across the growth stages and cultivars at the end of the flooding treatment, but there was no significant effect observed on dry adventitious root weight (Table 7). The application of auxin at a rate of 10 mg L$^{-1}$ gave the maximum fresh and dry adventitious root weights. This was most likely due to the highest number of adventitious roots produced per plant. In this study, the application of auxin at different growth stages had no significant effect on the dry adventitious root weight across auxin rates and cultivars at the end of the flooding treatment (Table 7). For sugarcane cultivars, K93-219 produced a significantly higher fresh adventitious root weight across auxin rates and growth stages at the end of the flooding treatment than the KK3 cultivar, but there was no significant difference in dry adventitious root weight (Table 7). However, the K93-219 cultivar tended to give a higher dry adventitious root weight than that of the KK3 cultivar. This was associated with K93-219 producing a higher number of adventitious roots than that of the KK3 cultivar. This result is in agreement with a previous study by [26] showing that under flooding stress, different varieties of sugarcane produced different numbers of adventitious roots. In the present study, there was a significant interaction between the effects of auxin rates and cultivars on the fresh adventitious root weight at the end of the flooding stress period and the number of adventitious roots at 7 DAT ($p \le 0.05$). The auxin application rates of 10, 20, and 30 mg L$^{-1}$ produced significantly different fresh adventitious root weights for both cultivars (Table 8). However, there were differences in the number of adventitious roots between auxin application at rates of 10, 20, and 30 mg L$^{-1}$ between the two sugarcane cultivars, but this was not observed in the no-auxin application (Table 8). A higher number of adventitious roots was exhibited by the K93-219 cultivar. Furthermore, there was a significant interaction between the effects of the growth stages and the cultivars observed in fresh adventitious root weights at the end of the flooding stress period ($p \le 0.05$). The application of auxin at 4 and 7 MOA had a significant effect on fresh adventitious root weights for both cultivars, but there was no significant effect observed at 5 MOA (Table 9). A higher fresh adventitious root weight was observed in the K93-219 cultivar.

**Table 8.** Interaction between auxin rates and two sugarcane cultivars in fresh adventitious root weights and adventitious root numbers at 7 days after treatment, Experiment 2.

| Auxin Rates (mg L$^{-1}$) (A) | Cultivars (C) | AR Fresh Weight (gm Plant$^{-1}$) | AR Number |
|---|---|---|---|
| 0 | KK3 | 15.4d | 78de |
| | K93-219 | 20.0bc | 89bcd |
| 10 | KK3 | 22.6ab | 92bc |
| | K93-219 | 27.2a | 113a |
| 20 | KK3 | 18.0bcd | 80cde |
| | K93-219 | 20.3b | 94b |
| 30 | KK3 | 15.1d | 72e |
| | K93-219 | 18.4bcd | 87bcd |
| F-test | | * | * |

Notes: * = significant at $p \le 0.05$. Mean with the difference, small letters in each column are significantly different by least significant difference ($p \le 0.05$); AR = adventitious root.

**Table 9.** Interaction between growth stages and two sugarcane cultivars in fresh adventitious root weights and adventitious root numbers at 7 days after treatment, Experiment 2.

| Growth Stages (G) | Cultivars (C) | AR Fresh Weight (gm Plant$^{-1}$) |
|---|---|---|
| 4 MOA | KK3 | 20.3c |
| | K93-219 | 28.5a |
| 5 MOA | KK3 | 20.1c |
| | K93-219 | 22.7bc |
| 7 MOA | KK3 | 20.8c |
| | K93-219 | 25.6b |
| F-test | | * |

Notes: * = significant at $p \leq 0.05$. Mean with the difference, small letters in each column are significantly different by least significant difference ($p \leq 0.05$); AR = adventitious root, MOA = month of age.

Foliar applied auxin can enter plants through the leaf epidermis or stomata and thereafter translocate via the apoplast or symplast pathways [56]. The xylem and phloem play important roles in the transport of auxin [57]. Foliar-sprayed auxin translocated down to the roots through the phloem [58]. The vacuole and cell wall serve as the main accumulation sites [59]. The efficiency of absorption, transport, and accumulation may be affected by abiotic factors (humidity and temperature) and plant physiological characteristics [60].

### 3.6. Aerenchyma Formation

Root aerenchyma formation in sugarcane is constitutive or induced by abiotic stresses such as hypoxia resulting from flood conditions [38]. According to [61] ethylene sensitivity in plants triggers root aerenchyma formation, while [62] reported that both ethylene sensitivity and ethylene-auxin balance may play a role in the formation of aerenchyma. Aerenchyma, a tissue comprising longitudinal channels filled with air by which the oxygen is transported from the atmosphere to the roots, was developed in the cortex part of the adventitious root during the flood period. The porosity of aerenchyma tissue in adventitious roots provides a better system of interconnected aerial spaces of lower resistance for oxygen transport from aerial shoots to submerged roots [63]. Based on the root cortex visual score by [39], the amount of aerenchyma formation was not significantly different ($p \leq 0.05$) at all auxin rate applications at different growth stages (Table 7). Aerenchyma was developed in the cortex of adventitious roots increasing their porosity under flooding conditions (Figure 3).

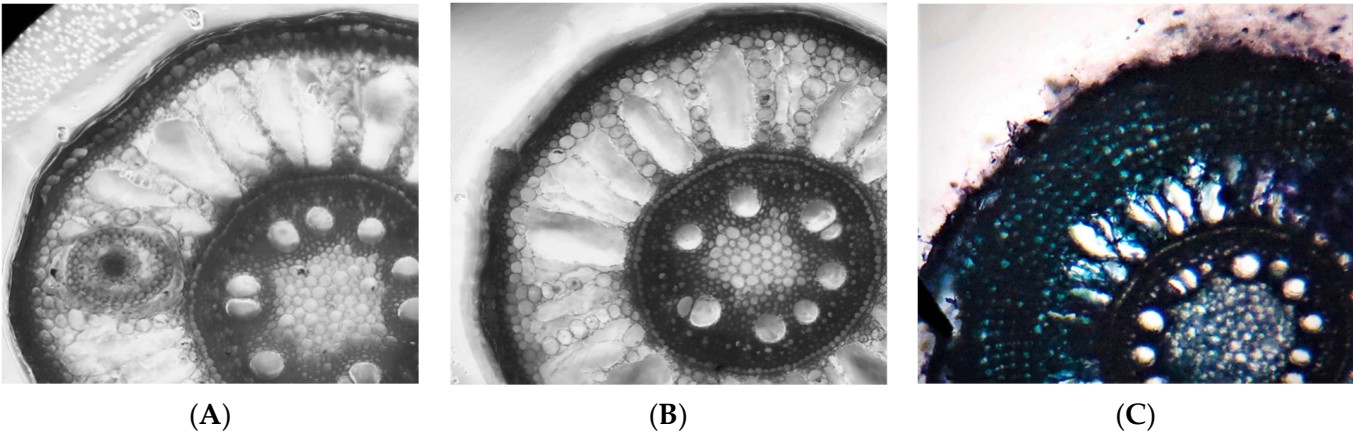

(**A**)  (**B**)  (**C**)

**Figure 3.** Formation of aerenchyma increased porosity in the cortex adventitious root during flood; Picture (**A**) (KK3) and picture (**B**) (K93-219) at 10 mg L$^{-1}$ of auxin application. Picture (**C**) indicates few aerenchyma formation of original root. Scale bars = 0.25 mm.

### 3.7. Growth Parameters and Stayed Green Leaf Performance

In the present study, auxin application rates had no significant effect on plant height, leaf width, number of stems, fresh stem weight, or leaves staying green across the growth stages for both cultivars at harvest (Table 10). The application of auxin at a rate of 10 mg $L^{-1}$ tended to increase fresh stem weight. The highest leaf width and fresh stem weight were observed in the auxin application made at 5 MOA. In the present study, both cultivars had the same leaf width, number of stems, fresh stem weight, and leaves staying green across auxin rates and growth stages (Table 10). However, the cultivar K93-219 was significantly taller than KK3 (Table 10). While auxin application at a rate of 10 mg $L^{-1}$ increased the number and weight of adventitious roots compared to other rates, including the control (0 mg $L^{-1}$), the plants did not exhibit significant differences in fresh stem weight. This was most likely due to flooding stress stimulating enough production of adventitious roots. Based on these observations, the external auxin application to promote adventitious root development is not useful.

**Table 10.** Growth parameters of two sugarcane cultivars as affected by auxin application rates at different growth stages at harvest, Experiment 2.

| Treatment | Plant Height (cm) | Stayed Green Leaf (No. Plant$^{-1}$) | Leaf Width (cm) | Stem Number (No. Plant$^{-1}$) | Stem Fresh Weight (kg Plant$^{-1}$) |
|---|---|---|---|---|---|
| Auxin Rates (A) | | | | | |
| 0 mg $L^{-1}$ | 132 | 26 | 4.7 | 3 | 5.8 |
| 10 mg $L^{-1}$ | 124 | 25 | 5.8 | 3 | 6.4 |
| 20 mg $L^{-1}$ | 123 | 26 | 6.7 | 3 | 6 |
| 30 mg $L^{-1}$ | 116 | 34 | 6.9 | 4 | 6 |
| F-test | Ns | Ns | ns | ns | ns |
| Growth stages (G) | | | | | |
| 4 MOA | 120 | 31 | 4.9c | 3 | 4.1b |
| 5 MOA | 124 | 25 | 7.1a | 3 | 5.9a |
| 7 MOA | 129 | 26 | 6.1b | 3 | 5.7a |
| F-test | Ns | Ns | * | ns | * |
| Cultivars (C) | | | | | |
| KK3 | 115b | 29 | 5.6 | 3 | 6.4 |
| K93-219 | 133a | 27 | 6.4 | 3 | 7.5 |
| F-test | * | Ns | ns | ns | ns |
| Interaction | | | | | |
| AxG | Ns | Ns | ns | ns | ns |
| AxC | Ns | Ns | ns | ns | ns |
| GxC | Ns | Ns | ns | ns | ns |
| AxGxC | Ns | Ns | ns | ns | ns |

Notes: ns, * = non-significant and significant at $p \leq 0.05$ probability levels, respectively. Mean with the difference, small letters in each column are significantly different by least significant difference ($p \leq 0.05$); MOA = month of age.

### 3.8. Sugar Quality

The auxin rates applied had no significant effect on Brix, polarity, purity, fiber, or CCS for either cultivars at harvest (Table 11). Similarly, the effect of the application made at different growth stages was significant for all sugar quality parameters (Table 11). Furthermore, the two sugarcane cultivars also had no significant effect on all of the sugar quality parameters at harvest (Table 11). In the present study, the Brix content values ranged from 18.8% to 19.8% despite undergoing flooding stress for 30 days. According to [26], Brix content ranges from 17.3% to 18.9% for 12 varieties exposed to flooding stress. This indicates that the Brix content of both sugarcane cultivars was maintained at a high level even if they underwent flooding stress.

**Table 11.** Sugar quality components of two sugarcane cultivars as affected by auxin application rates at different growth stages at harvest, Experiment 2.

| Treatment | Brix (%) | Polarity (%) | Purity (%) | Fiber (%) | CCS (%) |
|---|---|---|---|---|---|
| Auxin Rates (A) | | | | | |
| 0 mg L$^{-1}$ | 19.8 | 69.2 | 83.9 | 14.1 | 12.0 |
| 10 mg L$^{-1}$ | 19.2 | 64.4 | 82.2 | 14.7 | 10.8 |
| 20 mg L$^{-1}$ | 18.8 | 63.4 | 81.2 | 16.4 | 10.6 |
| 30 mg L$^{-1}$ | 19.2 | 66.2 | 80.4 | 26.0 | 11.1 |
| F-test | ns | Ns | Ns | ns | ns |
| Growth stages (G) | | | | | |
| 4 MOA | 19.1 | 64.9 | 81.4 | 16.5 | 11.2 |
| 5 MOA | 19.0 | 64.1 | 81.0 | 16.1 | 10.7 |
| 7 MOA | 19.7 | 68.5 | 83.3 | 13.8 | 11.8 |
| F-test | ns | Ns | Ns | ns | ns |
| Cultivars (C) | | | | | |
| KK3 | 19.2 | 65.5 | 82.1 | 15.0 | 11.1 |
| K93-219 | 19.4 | 66.1 | 81.7 | 20.6 | 11.4 |
| F-test | ns | Ns | Ns | ns | ns |
| Interaction | | | | | |
| AxG | ns | Ns | Ns | ns | ns |
| AxC | ns | Ns | Ns | ns | ns |
| GxC | ns | Ns | Ns | ns | ns |
| AxGxC | ns | Ns | Ns | ns | ns |

Notes: ns = non-significant at $p \leq 0.05$; MOA = month of age; CCS = Commercial Cane Sugar.

## 4. Conclusions

This study showed that the drought or flood conditions, whether alone or combined, did not affect sugarcane growth as indicated by the number of tillers, plant height, and sugar quality, but there was a significant difference in fresh stem weight at harvest. The maximum reduction of the fresh stem weight was observed in the drought (30 days) combined with flood (60 days) in the KK3 cultivar. The greatest reduction of fresh stem weight was noticed in the drought (30 days) combined with flood (30 days) in the K93-219 cultivar. However, between cultivars, KK3, being a drought-tolerant cultivar, tended to retain better growth than K93-219, especially for fresh stem weight. Physiological characters showed that stomatal conductance, chlorophyll content, and chlorophyll a fluorescence were not different among the water regime treatments, but under individual and the combination of prolonged drought, the growth performances were reduced. In addition, the K93-219 cultivars produced more adventitious roots than the KK3 cultivars under flood conditions, which may help in offsetting the reduced gas exchange during flooding. Sugar quality was not affected by different water regimes and cultivars in this study. However, the fresh stem weight of KK3 was higher than that of K93-219 under combined extreme drought and flooding stress conditions in the present study. Thus, planting cultivars that can tolerate these water stress conditions is a management approach that can help maintain sugarcane productivity in lowland or paddy field areas. The second experiment showed that the K93-219 cultivar, a flood-tolerant cultivar, had a higher response to auxin rates than the KK3 cultivar, a drought-tolerant cultivar. The morphological and growth parameters of sugarcane under flood conditions, which received foliar auxin, were not significantly different except for the plant height. The K93-219 cultivar produced a higher number of adventitious roots than the KK3 cultivar, especially in the presence of a 10 mg L$^{-1}$ application of auxin. However, the effect of timing applications of auxin at different growth stages was not demonstrated in this study. Similarly, the scores of aerenchyma formation among auxin application rates and timing were not different. Two sugarcane cultivars also showed the trend. After applied auxin, with differences of rates for two sugarcane cultivars, fresh stem weight was not different but showed significance differences in the growth stages. Similarly, sugar quality was not affected by auxin rates, application timing, and cultivar. Based on these outcomes, the application of external auxin is, in

general, not beneficial for sugarcane production under temporary flooding stress during the cropping season.

**Author Contributions:** Conceptualization, A.P. (Anan Polthanee); methodology, A.P. (Anan Polthanee), V.T.-l. and J.B.; formal analysis, A.P. (Arunee Promkhambut) and J.B.; investigation, A.P. (Anan Polthanee); resources, J.B.; data curation, J.B.; writing—original draft preparation, writing—review and editing, J.B., A.P. (Anan Polthanee) and B.T.; supervision, A.P. (Anan Polthanee). All authors have read and agreed to the published version of the manuscript.

**Funding:** This research was funded by the Thai Royal Golden Jubilee Ph.D. Program (Grant no. PHD/0071/2556).

**Institutional Review Board Statement:** Not applicable.

**Informed Consent Statement:** Not applicable.

**Data Availability Statement:** Not applicable.

**Acknowledgments:** We would like to express our gratitude to the Thai Royal Golden Jubilee Program (Grant no. PHD/0071/2556) and Research Scholarships organized by Graduate School, Khon Kaen University, Thailand.

**Conflicts of Interest:** No conflicts of interest are declared.

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
