# Peer review of "Effects of Water Stress and Auxin Application on Growth and Yield of Two Sugarcane Cultivars under Greenhouse Conditions"

_agriculture, doi:10.3390/agriculture11070613_

Round 1

Reviewer 1 Report

The manuscript needs substantial improvement and language editing, as some parts are unclear or difficult to follow. Moreover, some parts give the impression that they were written by a less experienced scientist (e.g. a graduate student), compared with other parts that indicate a more experienced writer. These parts are not necessarily different in their scientific quality, but they likely indicate variable ability of authors to produce sophisticated text.

Citation of references is completely confusing. (starts with 4?)

Introduction is very weak, up to date references missing - only one from 2019 and 2020. I recommended add: 10.3390/plants9080931; 10.1093/pcp/pcz229; 10.1186/s12870-018-1428-9; 10.1111/tpj.15276; 10.1038/s41598-020-63006-7; 10.1007/s00299-021-02683-8;   10.3390/plants9121735

Better change tables to figs, variance is missing, HSD (Tukey) or other right statistical test also, eg. tab. 10-11 - there are any significant results?

Add more information on the absorption pathways (vacuolar) of plants under stress (water, drought). The discussion section needs to be revised. Arguments require a clearer and more accurate presentation. The understanding of plant uptake mechanisms is limited because it is limited to works that have a specific view and deliberately ignore alternatives, and do not represent a balanced view of the evidence.

Reviewer 2 Report

I found the manuscript of Jiraporn Bamrungrai and his colleagues very interesting. The authors evaluate the individual and combined effects of drought and flooding on two sugarcane varieties and the effect of auxin doses at different growth stages on two sugarcane varieties under flooding conditions.

This research has important implications for regional agriculture and provides new insights into the effects of abiotic stressors on sugarcane physiological parameters.

The manuscript is clear, well written and a sound experimental and statistical approach has been used.

Some specific comments are as follows:

1) What is meant by the photosynthetic activity index (L. 146)?

2) L. 135 corect mg kg-1

3) Stomatal conductance was measured using a leaf porometer SC-1 (Decagon De-142 vices, Inc., USA). In what units are the values given? Please add to the text.

4) Fill in the notes below the tables: different letters indicate a significant difference.

5) Figure 4: add significant differences using the letters for cultivars (Original root weight).

6) Check and standardize the style of references according to the journal requirements.

Reviewer 3 Report

The authors have investigated the effects of water stress and auxin application on physiological characters, growth and sugar quality of two sugarcane cultivars, KK3 (drought tolerant) and K93-219 (flood tolerant).

According to the water stress issue that limits the growth and productivity of sugarcane in Thailand, the authors employed nine water regimes. However, this factor was dominant when analyzing data, values for each cultivar under individual treatment was not shown and statistically analyzed. For each set of data, the results of KK3 and those of K93-219 should be separated, and results of each one of eight water stress regimes should be compared with the results of control respectively. After re-analyzing data, different conclusions may come out. It would be a good idea to tailor results for highlighting certain points of the study.

Additionally, there are a number of areas in which the paper can be improved.

  1. The labels of treatment in Figure 1 (T1-T9) are different from those in other tables (Control, Drought D1, Prolonged Drought D2, etc.).
  2. Is there data for the yield of CCS?
  3. At the end, the conclusion was addressed as “K93-219 has higher potential in surviving in flood condition than KK3 cultivar”, which is already known. It’d better to reconsider the conclusions and main contributions of the paper.

Round 2

Reviewer 1 Report

Authors not adressed all comments of reviewer. Manuscript still needs corrections and improvements. Still missing up to date references - i recommended some in previous review.

Fig. 3 - scale still missing,

Reviewer 3 Report

Authors have improved the manuscript markedly. This study will provide fundamental data on the effects of water stress and auxin application on the two sugarcane cultivars, KK3 and K93-219.

Author Response

Response to Academic Editor

Dear Editor/Reviewers

We appreciate the valuable suggestions for further improvement of the manuscript no. Agriculture-1239297 entitled " Effects of Water Stress and Auxin Application on Growth and Yield of Two Sugarcane Cultivars under Greenhouse Conditions". The changes in the manuscripts are made according to the comments and suggestions of the reviewers and Editor. The changes in the manuscript are indicated by "Track changes". The details of the revision are given below.

Best Regards

Anan Polthanee

Reviewer 3

Comments and Suggestions for Authors

Authors have improved the manuscript markedly. This study will provide fundamental data on the effects of water stress and auxin application on the two sugarcane cultivars, KK3 and K93-219.

  • Thank you very much for your kind suggestions.